# Comparative Genomics Reveal Phylogenetic Relationship and Chromosomal Evolutionary Events of Eight Cervidae Species

**DOI:** 10.3390/ani14071063

**Published:** 2024-03-30

**Authors:** Lixin Tang, Shiwu Dong, Xiumei Xing

**Affiliations:** State Key Laboratory for Molecular Biology of Special Economic Animals, Institute of Special Animal and Plant Sciences, Chinese Academy of Agricultural Sciences, Changchun 130112, China; tanglixin1217@163.com (L.T.); dongshiwu2017@126.com (S.D.)

**Keywords:** comparative genomic, Cervidae, phylogenetic relationship, chromosomal evolutionary events

## Abstract

**Simple Summary:**

Cervidae represents a substantial family within the Artiodactyla, yet their phylogenetic relationship has not been fully clarified due to limited data available. With the increasing availability of genomic data, there is now an opportunity to further explore their phylogenetic relationship. In our study, we employed a comparative genomics approach focusing on the chromosome-level genomes of eight Cervidae species to explore their phylogenetic relationship and chromosomal evolutionary events. Furthermore, we conducted an analysis of gene families to identify those potentially involved in adaptive evolution.

**Abstract:**

Cervidae represents a family that is not only rich in species diversity but also exhibits a wide range of karyotypes. The controversies regarding the phylogeny and classification of Cervidae still persist. The flourishing development of the genomic era has made it possible to address these issues at the genomic level. Here, the genomes of nine species were used to explore the phylogeny and chromosomal evolutionary events of Cervidae. By conducting whole-genome comparisons, we identified single-copy orthologous genes across the nine species and constructed a phylogenetic tree based on the single-copy orthologous genes sequences, providing new insights into the phylogeny of Cervidae, particularly the phylogenetic relationship among sika deer, red deer, wapiti and Tarim red deer. Gene family analysis revealed contractions in the olfactory receptor gene family and expansions in the histone gene family across eight Cervidae species. Furthermore, synteny analysis was used to explore the chromosomal evolutionary events of Cervidae species, revealing six chromosomal fissions during the evolutionary process from Bovidae to Cervidae. Notably, specific chromosomal fusion events were found in four species of Cervus, and a unique chromosomal fusion event was identified in *Muntiacus reevesi*. Our study further completed the phylogenetic relationship within the Cervidae and demonstrated the feasibility of inferring species phylogeny at the whole-genome level. Additionally, our findings on gene family evolution and the chromosomal evolutionary events in eight Cervidae species lay a foundation for comprehensive research of the evolution of Cervidae.

## 1. Introduction

Cervidae is one family of artiodactyls with abundant species [1]. The wide geographical distribution, diverse body size and habitats make the identification of phylogenetic relationships difficult. The phylogeny of Cervidae has been extensively studied, including morphology, mitochondrial genomes and mitochondrial marker sequences [2,3,4]. The consensus for phylogeny of Cervidae can only be partially reached.

Sika deer (*Cervus nippon* Temminck, 1838) and red deer (*Cervus elaphus* Linnaeus, 1758) within Cervus are closely related in term of evolutionary relationship, yet their phylogenetic relationship has been controversial. Studies based on mitochondrial genomes have classified red deer (*Cervus elaphus*) [5], Tarim red deer (*Cervus hanglu*) [6] and wapiti (*Cervus canadensis*) [7] as three separate species [8]. The latest classification has provided a basis for a more accurate study of the phylogenetic relationship between sika deer (*Cervus nippon* Temminck,1838) and red deer (*Cervus elaphus* Linnaeus, 1758) as well as the phylogenetic relationship among red deer (*Cervus elaphus*), wapiti (*Cervus canadensis*) and Tarim red deer (*Cervus hanglu*). Mitochondrial genome studies have indicated that there are western and eastern mtDNA lineages, with red deer and Tarim red deer in the western lineage, with wapiti in the eastern lineage and closely related to sika deer [9,10]. However, there is research that suggests that wapiti belong to the eastern lineage and are closely related to sika deer, while red deer belong to the western lineage. Additionally, the Tarim red deer is a separate lineage, showing a closer association with the red deer, indicating that wapiti and sika deer may have a common ancestor that is separated from red deer [11]. These studies based on mitochondrial genomes and mitochondrial marker sequences cannot provide a uniform conclusion on the phylogeny of Cervus.

The karyotypes of Cervidae exhibit significant diversity in diploid chromosome number (2n = 6–70) [12], indicating abundant chromosomal evolution events of Cervidae species. With the vigorous development of sequencing technology, an increasing number of species now possess chromosome-level genomes, enabling the exploration of chromosome evolution at the genomic level [13,14,15,16]. Comparative genomics is extensively utilized to explore evolutionary relationships and chromosome evolution among different species and large populations [17,18]. In this study, we utilize a comparative genomic approach, focusing on eight high-quality genomes of Cervidae, to explore the phylogenetic relationship at the genomic level. Additionally, our aim is to leverage chromosome-level genomes to elucidate the chromosomal evolutionary events of Cervidae.

## 2. Materials and Methods

### 2.1. Genome Data

The genome assemblies of nine mammalian species were used in this study and these chromosome-level genomes were downloaded from NCBI (https://www.ncbi.nlm.nih.gov/assembly (accessed on 12 December 2022)) and NGDC (https://ngdc.cncb.ac.cn/gwh/ (accessed on 12 December 2022)) (Table 1). There were eight species of Cervidae included, including sika deer (*Cervus nippon*, mhl_v1.0), red deer (*Cervus elaphus*, mCerEla1.1), Tarim red deer (in press), wapiti (*Cervus canadensis*, ASM1932006v1), reindeer (*Rangifer tarandus*, mRanTar1.h1.1), white-tailed deer (*Odocoileus virginianus*, Ovir.te_1.0), Reeves’ muntjac (*Muntiacus reevesi*, ASM2022604v1) and red muntjac (*Muntiacus muntjac*, UCB_Mmun_1.0). In addition, cattle (*Bos taurus*, ARS-UCD2.0) was used as the outgroup.

### 2.2. Phylogenetic Analysis

The phylogenetic tree was constructed using single-copy orthologous gene sequences. OrthoFinder v2.5.4 [19] was used to identify single-copy orthologous genes. SeqKit v0.15.0 [20] was used to extract the CDS of single-copy orthologous genes from the genomes of nine species. MUSCLE v3.8.31 [21] was used for aligning the CDS of single-copy orthologous genes. In order to ensure the sequences of different species remained in the same order in the files after aligning, SeqKit v0.15.0 was used to sort the aligned sequences. Finally, the sorted sequences were spliced manually, trimmed using TrimAl v1.2 [22] and converted to PHYLIP format. The phylogenetic tree was constructed using RAxML- v8.2.12 [23] with the GTRCAT model and cattle serving as the outgroup. 

**Table 1 animals-14-01063-t001:** The genomic information of nine species used in our study.

Subfamily or Tribe	Species	Genome Size (Mb)	Scaffold N50 (Mb)	Haploid Karyotype
Cervini	Sika deer [24] (*Cervus nippon*)	2500.64	78,786,809	32 + X
	Red deer [25] (*Cervus elaphus*)	2886.60	83,473,711	33 + X
	Tarim red deer (in press) (*Cervus hanglu*)	2520.87	76.801.786	33 + X
	Wapiti [26] (*Cervus canadensis*)	2526.61	77,654,944	33 + XY
Capreolinae	Reindeer (*Rangifer tarandus*)	2971.15	69,829,200	34 + XY
	White-tailed deer [27] (*Odocoileus virginianus*)	2380.49	-	-
Muntiacini	Reeves’ muntjac [28] (*Muntiacus reevesi*)	2494.01	113,316,579	22 + X
	Red muntjac [29] (*Muntiacus muntjak*)	2489.50	682,452,208	2 + X
Bovinae	Cattle [30] (*Bos taurus*)	2770.67	103,308,737	29 + XY

The genome of white-tailed deer is not at the chromosome level.

### 2.3. Divergence Time Estimation

Three calibration time points, namely Bovidae vs. Cervidae (18.5~27.8 Mya), reindeer vs. white-tailed deer (5.6~11.4 Mya) and Reeves’ muntjac vs. red muntjac (2.5~9 Mya) were chosen from TIMETREE (http://timetree.org/ (accessed on 5 May 2023)) [31]. The adjusted calibration time points were 23.7 Mya, 7.8 Mya and 4.94 Mya, respectively. The adjusted calibration time points were used to estimate the evolutionary rates of single-copy gene sequences of eight species by MCMCtree in PAML v8.2.12 [32]. The divergence times of eight Cervidae species were estimated using the evolutionary rates and the phylogenetic tree we obtained.

### 2.4. Evolution of Gene Families

Gene family expansion and contraction were performed using CAFE v5.0 [33]. The gene families were identified using OrthoFinder v2.5.4 and gene families with large copy number differences between different species were eliminated. The phylogenetic tree with time-calibrated was used as the input data. The expanded/contracted gene family with a *p*-value ≤ 0.01 was defined as a “significantly expanded/contracted gene family”.

### 2.5. Chromosome Evolution

To identify synteny blocks among the chromosome-level genomes of eight species, the All-vs.-All blastp with e-value < e × 10^−5^ was conducted for protein genomes of each genome pair. The synteny blocks were scanned using MCScanX [34] with default settings except for “gap_penalty -3”. The chromosome-scale syntenies between species were visualized by NGenomeSyn v1.41 [35].

### 2.6. Gene Enrichment Analysis

To further elucidate the biological functions of significantly contracted and expanded genes in eight Cervidae species, functional enrichment analyses were performed. The gene annotations for GO [36] and KEGG [37] were accomplished using Swiss-Prot (https://ftp.uniprot.org/pub/databases/uniprot/current_release/knowledgebase/complete/uniprot_sprot.fasta.gz, accessed on 24 February 2024) and KOBAS v3.0. The enrichment analyses were performed using clusterProfiler v4.8.2 [38].

## 3. Results and Discussions

### 3.1. Phylogenetic Analysis

The available genome data for eight Cervidae species were downloaded for phylogenetic analysis, with cattle serving as an outgroup (Table 1). A phylogenetic tree was constructed with 2480 single-copy orthologous genes from nine species (Figure 1a, Appendix A). The relationship of these species can be seen in that the Cervidae are divided into two main branches: one is the Capreolinae, and the other is Cervinae, which contains two tribes, the Muntiacini and Cervini. Previous studies on the phylogeny of Cervidae based on morphology, mitochondrial genomes and mitochondrial marker sequences have consistently classified the family mainly into two subfamilies, Capreolinae and Cervinae, corresponding to Telemetacarpalia and Plesiometacarpalia, respectively [39]. In our study, we conducted phylogenetic analysis using the whole genomes of eight Cervidae species. Although our samples are relatively limited, we have to cover as many individuals from each subfamily of the Cervidae as possible in our selection. These are the genomes of Cervidae that are currently available. Moreover, our phylogeny result of Cervidae is consistent with previous research [9,40]. While there may be limitations in using individuals instead of populations, the findings of our study are generally in line with those of population-based studies [41,42]. This is because whole genomes contain a significant amount of genetic information, which is sufficient to reflect the evolutionary relationship between species. Additionally, the genomic differences between individuals can also indicate the genetic diversity within a population, providing a reliable foundation for studying the evolutionary relationship between species. Indeed, research conducted on various species has validated the efficacy of utilizing individual genomes to uncover evolutionary relationships [43,44].

Notably, the four species in Cervus include red deer, Tarim red deer, wapiti and sika deer. It can be observed from the phylogenetic tree that the Tarim red deer shares a branch with the sika deer, while the red deer and wapiti form another branch. Tarim red deer is more closely related to sika deer than to red deer and wapiti. The evolutionary relationship among sika deer, wapiti, Tarim red deer and red deer have been controversial [8,11,45]. Previous studies based on complete mitochondrial genomes have generally suggested that wapiti are more closely related to sika deer than to red deer and Tarim red deer [8,9,46]. However, our study, which utilized single-copy orthologous genes to construct the phylogenetic tree, indicates that Tarim red deer are more closely related to sika deer than to red deer and wapiti. This discrepancy in results may be attributed to the limitations of mitochondrial genome analysis, which only represents maternal inheritance. In contrast, our study utilizes the whole genome sequence, encompassing a vast number of nuclear and cytoplasmic genes that have undergone distinct evolutionary processes [47]. Additionally, the whole-genome single-copy orthologous genes are widely used in the study of phylogenetic relationships among species [48,49]. By considering these factors, the evolutionary relationships between species can be more accurately revealed using whole-genome data [50].

### 3.2. Divergence Time Estimating

Divergence times were estimated using MCMCtree in PAML v4.8 [32]. The common ancestor of Cervidae and Bovidae is identified about 21.2 million years ago. Capreolinae separated from Cervidae about 14.2 million years ago. The divergence time between Muntiacini and Cervini was around 12.57 million years. The four species in Cervus were divided into two branches, with the divergence from a common ancestor of the two branches occurring approximately 3.35 million years ago. The divergence time between Tarim red deer and sika deer was around 3.1 million years and the divergence time between red deer and wapiti was around 1.9 million years (Appendix A).

The divergence times of each subfamily of Cervidae obtained in our study are different from those previously obtained based on mitochondrial genomes and mitochondrial makers [8,9]. This difference disparity can be attributed to the types of data used [51], estimation methods employed [52] and the choice of fossil calibration points [53]. All of these factors have an impact on the estimation of species divergence time. The method used in this study for estimating divergence time has been widely used to estimate species divergence time based on whole-genome data [54,55,56]. Additionally, the fossil calibration points used to estimate the divergence time were obtained from TIMETREE (http://timetree.org/ (accessed on 5 May 2023)) [57], which is also used for many other species to obtain fossil calibration points [58]. To sum up, the differences of divergence time in our study are within reasonable limits.

### 3.3. Gene Family Evolution

There are 28,003 orthogroups (gene family clusters) that were identified by OrthoFinder v 2.5.4 [19], of which 26,348 comprised two or more species (Appendix A) and 12,978 were present in all species (Figure 1b, Appendix A). Among these, 2480 orthogroups consisted entirely of single-copy genes with a one-to-one relationship in different genomes (Appendix A). Additionally, the species-specific orthogroups of the eight species were also identified (Appendix A). To understand adaptive evolution gene families among the eight species of Cervidae, an analysis of gene family evolution was performed using CAFE v5.0 [33] (Figure 1a). The number of significant contracted and expanded gene families (*p* < 0.05) in these species were counted (Table 2). Furthermore, we performed GO [59] and KEGG analysis [60] to explore the function involved in significantly contracted and expanded gene families.

The expansion and contraction of gene families are common phenomena in the genome evolution of species and play an important role in speciation, adaptation and genome stability [61,62]. Our results showed that the significant contracted gene family among eight Cervidae species is the olfactory receptor family, which involves biological functions and signaling pathways including odorant binding (GO:0005549, *p* < 0.01), sensory perception of smell (GO:0007608, *p* < 0.01) and olfactory transduction (ko04740, *p* < 0.01) (Appendix A). In mammalian species, the olfactory system plays a crucial role in mate detection [63], risk avoidance [64] and other survival functions. However, the number of olfactory receptors changes as species adapt to the environment [65]. The significant contraction of olfactory receptors in eight Cervidae species may be attributed to the habitat destruction and habitat fragmentation as well as increasing domestication that led to stable and single living environments [66,67]. Additionally, artificial interference in their search for mates and foraging may also contribute to the loss of olfactory receptor genes [68,69]. The adaptation of Cervidae to the changed environment results in the contraction of olfactory receptors. A previous study also confirmed the adaptive evolution of olfactory-related genes in Cervidae species [70].

The significant expanded gene family is the histone gene family (*H2A*, *H2B*, *H3* and *H4*), which involves biological functions and signaling pathways including structural constituent of chromatin (GO:0030527, *p* < 0.01) and nucleosome (GO:0000786, *p* < 0.01) (Appendix A). Histones are highly conserved in mammals and play a central role in transcription regulation, DNA repair, DNA replication and chromosomal stability through acetylation, phosphorylation, methylation and ubiquitination [71].

The specific expansion gene family of sika deer is the keratin gene family (*KRTHB2, KRTHB4, KRTHB1, KRTHB3*), which involves biological functions and signaling pathways including structural constituent of skin epidermis (GO:0030280, *p* < 0.01) and keratinization (GO:0031424, *p* < 0.01). In mammals, keratins are key components of the cytoskeleton, providing mechanical stability to the epidermis. Their role is crucial for the skin as a barrier against the external environment [72]. The keratin gene family has been found to be related not only to animal hair color but also to animals’ ability of maintain body temperature [73,74]. Comparative genomics studies of terrestrial and whole-aquatic mammals have found that the number of keratin gene families in animals living in different environments changes in order to adapt to the environment [75]. Sika deer are primarily distributed in the cold environment of northeastern China [68]. The significant expansion of this gene family may be associated with the evolutionary adaptation to the cold environment.

### 3.4. Chromosome Evolution of Cervidae

The chromosome evolution events occurring in eight species of Cervidae (*Rangifer tarandus* (RTA), *Cervus nippon* (CNI), *Cervus elaphus yarkandensis* (CELY), *Cervus canadensis* (CCA), *Cervus elaphus* (CEL), *Muntiacus reevesi* (MRE), *Muntiacus muntjak* (MMU)) were assessed. To discuss chromosome dynamics of these species, *Bos taurus* (BTA) was used as a reference and the species’ order of chromosome collinearity corresponded to the position in the phylogenetic tree in Section 3.1. Corresponding chromosomes and chromosome evolution events can be easily traced in Figure 2.

According to the previous report, the segments of the last common ancestor of Cervidae and *Bos taurus* correspond to the two cattle chromosomes BTA26 and BTA28 [76]. The two chromosomes were fused and presented as a single chromosome in Cervidae (BTA26 and BTA28 → RTA7 → CNI9 →CELY9→ CCA8→ CEL15 → MRE chr2 → MMU chr2) (Appendix A).

Karyotypes of Capreolinae are conserved and predominantly represented by an ancestral karyotype (2n = 70) [12]. Reindeer, closely related to the ancestral karyotype of Cervidae, has retained the karyotype (2n = 70). From the collinearity result of *Bos taurus* and *Rangifer tarandus*, we can see that twelve chromosomes of reindeer arose by fission of six cattle chromosomes (BTA1 → RTA5 and RTA29; BTA5 → RTA25 and RTA27; BTA2 → RTA13 and RTA24; BTA8 → RTA18 and RTA30; BTA9 → RTA22 and RTA34; BTA6 → RTA17 and RTA32) (Appendix A).

The chromosomal collinearity of BTA17 and BTA19 correspond to RTA15 and RTA21. However, in four species of Cervus, two chromosomes fused as a chromosome (BTA17 and BTA19 → CNI2 → CELY2 → CCA1 → CEL5) (Appendix A); this fusion is unique in the Cervus lineage.

Among the four species in Cervus, the chromosome number of *Cervus elaphus* (CEL), *Cervus elaphus yarkandensis* (CELY) and *Cervus canadensis* (CCA) is 2n = 68, while *Cervus nippon* (CNI) has a chromosome number of 2n = 66. The collinearity results show that one chromosome of *Cervus nippon* corresponds to two chromosomes of the other three species (CNI1 → CELY18 and CELY11 → CCA18 and CCA10 → CEL4 and CEL23) (Appendix A). However, it is currently uncertain in our study whether the chromosomal evolutionary event among the four species is chromosome fission or chromosome fusion. Nevertheless, some studies suggested that the chromosomal evolutionary pattern between sika deer and red deer was chromosome fission, which means the divergence time of sika deer is earlier than that of red deer [77]. And a study based on cytogenetics has suggested that chromosome fusion played a great role in karyotypic differentiation in Cervinae [78]. It is necessary to obtain as much chromosome-level genome data as possible and combine them with cytogenetic data to revolve the chromosomal evolutionary events of Cervus.

Species of Muntiacini exhibit remarkable variations in their chromosomal karyotypes, particularly in terms of the number of chromosomes. These variations range from 2n = 6 in the female *Muntiacus muntjak* to 2n = 46 in the *Muntiacus reevesi*. Previous research has suggested that *Muntiacus reevesi* is the more primitive species within the Muntiacini [79,80]. Chromosome fusion and fusion were considered to be the major chromosomal evolutionary events of Muntiacini [79,81,82]. However, our study found a specific fusion of *Muntiacus reevesi* (BTA29 and BTA16 → RTA26 and RTA9 → CNI28 and CNI13 → CELY29 and CELY14 → CCA29 and CCA3 → CEL2 and CEL14→ MRE5 → MMU1 and MMU3X) (Appendix A). Furthermore, our result indicated that the chromosomal evolutionary events in *Muntiacus reevesi* and *Muntiacus muntjak* were mainly chromosome fusion, which is consistent with previous studies [28,29,83].

## 4. Conclusions

Overall, our study, using eight high-quality genomes of Cervidae as research objects, provides new insights into the phylogeny of Cervidae, particularly the phylogenetic relationships among sika deer, red deer, wapiti and Tarim red deer. Meanwhile, we preliminarily explored the chromosomal evolutionary events of the eight Cervidae species. Gene family evolution analysis revealed that olfactory receptor gene family has contracted, while the histone gene family has expanded in eight Cervidae species. This study demonstrates the feasibility of resolving phylogenetic problems using whole genomes. To more comprehensively explore the phylogenetic relationships and the chromosomal evolutionary events of Cervidae, the chromosome-level genomes of Cervidae will be needed. On the basis of cytogenetic studies, future research will focus on understanding the characteristics of chromosomal evolutionary events in genome sequences and their impacts on genomes.

## Figures and Tables

**Figure 1 animals-14-01063-f001:**
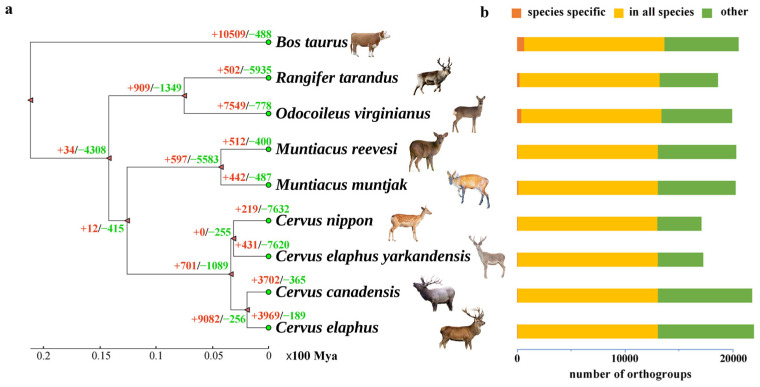
Phylogeny and gene family analysis of Cervidae. (**a**) Phylogeny with divergence time and gene family expansion and contraction in the eight Cervidae species, with *Bos taurus* used as an outgroup. The numbers of expanded (red) and contracted (green) gene families are shown on branches. (**b**) Horizontal bar plots indicate the number of orthogroups that are species-specific (red), present in all nine species (orange), or present in more than one but less than all species (green) in the analysis.

**Figure 2 animals-14-01063-f002:**
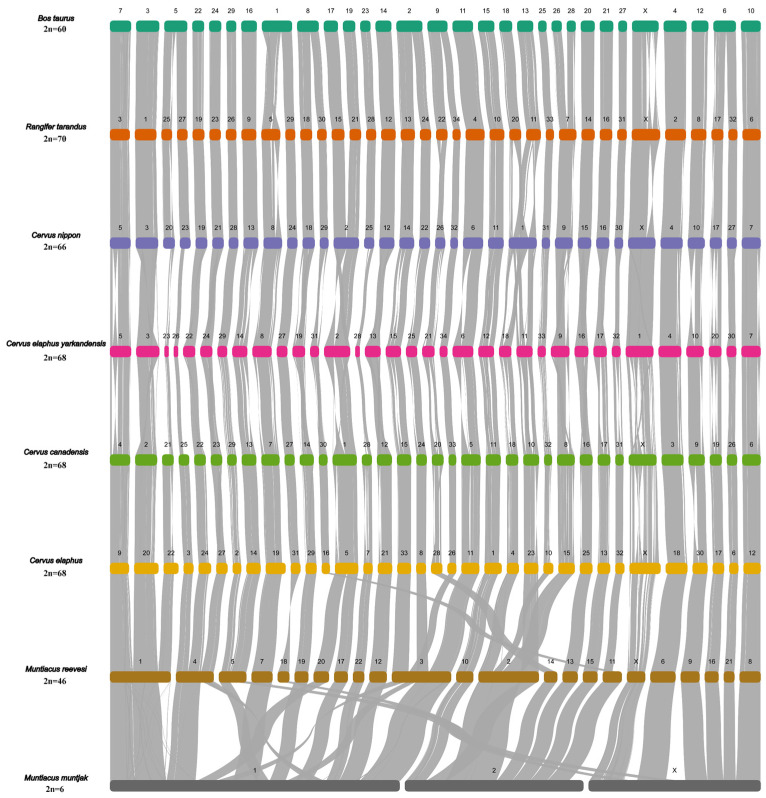
The chromosome collinearity of seven species of Cervidae (*Cervus elaphus yarkandensis*, *Cervus nippon*, *Cervus elaphus*, *Cervus canadensis*, *Rangifer tarandus*, *Muntiacus reevesi* and *Muntiacus muntjac*), with *Bos taurus* as the reference. *Odocoileus virginianus* was excluded, as it is not a chromosome-level genome. The cirves are in the same colors, grey. The different colors are the chromosomes of different species.

**Table 2 animals-14-01063-t002:** Statistics on significantly contracted/expanded gene families/genes in eight species.

Species	Contractive		Expansive	
	Gene Families	Genes	Gene Families	Genes
Reindeer	1795	1917	108	1434
White-tailed deer	162	273	1690	12,418
Reeves’ muntjac	136	365	126	1285
Red muntjac	138	315	126	1230
Red deer	32	70	2109	19,456
Wapiti	67	144	1994	18,075
Tarim red deer	2212	2121	25	202
Sika deer	2202	2127	21	286

## Data Availability

Data are contained within the article and Appendix A.

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
