# Peer review of "Comparative Genomics Reveal Phylogenetic Relationship and Chromosomal Evolutionary Events of Eight Cervidae Species"

_animals, 2024, doi:10.3390/ani14071063_

Round 1

Reviewer 1 Report

Comments and Suggestions for Authors

Cervidae family includes several species with wide geographical distribution, diverse body size and habitats. Still, some controversy exists regarding phylogenetic relationship of the species under Cervidae. Although, mitochondrial genome and mitochondrial marker based approaches indicated that Cervidae is divided into four subfamilies, Odocoileinae, Hydropotinae, Muntiacinae and Cervinae, a broad consensus on this has not been established. Therefore, the authors employed a comparative genomic approach using eight high-quality genomes of Cervidae to address the issue. Moreover, the chromosomal evolutionary events of different subfamilies were also elucidated. The objective of the study is very clear, the authors identified a gap in existing knowledge and addressed it.

1.      Introduction is clear and to the point.

2.      M & M includes sufficient details.

3.      The results have been depicted nicely.

 Overall, the work is very important, rigorous and well presented. The writing is also very good. I do not have any concern with the work.    

Comments on the Quality of English Language

Moderate editing is required.

Author Response

Thank you for your comments, we have refined the article.

Reviewer 2 Report

Comments and Suggestions for Authors

The article is potentially interesting, but I think that it needs a major revision. I am a zoologist who is working on cervid taxonomy and evolution, therefore I cannot give a qualified assessment on methodology and used data, however, there are several points that are under my expertise and here are my suggestions concerning those points:

General Comment: The authors acknowledge discrepancies between their results and previous studies regarding the phylogenetic relationships among Cervus elaphus, Cervus canadensis, and Cervus nippon. I suspect that the issue lies in the taxonomic assignment of the samples. While the authors mention the subspecies of red deer from the Tarym area (Cervus elaphus yarkandensis), they fail to specify the subspecies of other "Cervus elaphus" samples and their geographical origins.  Is this sample coming from Europe or Balkan-Caucasian-Caspian area? In this case it should be Cervus elaphus elaphus or Cervus elaphus maral correspondingly. If the sample is coming from Siberia, it could be a source of misunderstanding. Siberian deer are sometimes misidentified as a subspecies of red deer Cervus elaphus sibiricus, when they are actually Asian wapiti Cervus canadensis sibiricus. I recommend verifying and specifying the subspecific identity and geographic provenance of the "Cervus elaphus" sample used in the study.

Page 1, lines 40-42: “The study based on mitochondrial genome and mitochondrial marker sequences show that Cervidae is divided into four subfamilies, Odocoileinae, Hydropotinae, Muntiacinae and Cervinae[3-5]” – the citations are not exact. Currently, there are only two modern monophyletic cervid subfamilies recognized: the subfamily Capreolinae (Odocouleinae is its junior synonym; this group is also known as telemetacarpal deer) and the subfamily Cervinae (or plesiometacarpal deer). Hydropotes is closely related to Capreolus and is included in the subfamily Capreolinae by all researchers. There are no debates concersisn the systematical position of Hydropotes in the current literature.

Page 2, line 50: "Cattle" should be capitalized.

Page 3-4, lines 120-121: “corresponding to Telemetacarpalia and Plesiometacarpalia, respectively, which also proposed by Brooke [23]” - better to move this sentence on correspondence of Capreolinae and Cervinae to Brooke's Telemetacarpalia and Plesiometacarpalia to the introduction chapter.

Page 4, lines 123-125: “However, previous studies have indicated that Hydropotes inermis is a distinct subfamily called Hydropotinae, which belongs to the Telemetacarpalia[24,25]” - The citations are not exact. Randi et al (1998) arrived to conclusion that Hydropotes is nested within the Odocoileinae, i.e. belongs to the subfamily Caprelinae. Gilbert et al (2006) included Hydropotes together with Capreolus to the tribe Capreolini within the subfamily Capreolinae.

Page 8, line 277: “Adaptive evolution in olfactory receptor gene families of eight Cervidae species were discovered” – The statement regarding the discovery of adaptive evolution in olfactory receptor gene families among eight Cervidae species requires further elaboration. It would be beneficial to include details such as the specific adaptations observed, the environmental conditions influencing these adaptations, the selection pressures involved, and the advantages conferred by these adaptations. Providing a summarized information and conclusion on these discoveries would enhance the clarity and impact of the study.

Author Response

1.General Comment: The authors acknowledge discrepancies between their results and previous studies regarding the phylogenetic relationships among Cervus elaphus, Cervus canadensis, and Cervus nippon. I suspect that the issue lies in the taxonomic assignment of the samples. While the authors mention the subspecies of red deer from the Tarym area (Cervus elaphus yarkandensis), they fail to specify the subspecies of other "Cervus elaphus" samples and their geographical origins.  Is this sample coming from Europe or Balkan-Caucasian-Caspian area? In this case it should be Cervus elaphus elaphus or Cervus elaphus maral correspondingly. If the sample is coming from Siberia, it could be a source of misunderstanding. Siberian deer are sometimes misidentified as a subspecies of red deer Cervus elaphus sibiricus, when they are actually Asian wapiti Cervus canadensis sibiricus. I recommend verifying and specifying the subspecific identity and geographic provenance of the "Cervus elaphus" sample used in the study.

Thanks for your comments. In order to resolve confusion about sample information, we have added the reference for every sample in Table 1. And we would like to answer you the question about Cervus elaphus and Cervus canadensis in our study.

The “Cervus elaphus” in our study is Cervus elaphus hippelaphus, the Central-European red deer. The sample was obtained at the Deer Farm of the Game Management and Landscape Center of Kaposvár University (Bőszénfa, Hungary). You can get some details in the reference “Bana, N.Á., Nyiri, A., Nagy, J. et al. The red deer Cervus elaphus genome CerEla1.0: sequencing, annotating, genes, and chromosomes. Mol Genet Genomics 293, 665–684 (2018). https://doi.org/10.1007/s00438-017-1412-3”.

The “Cervus canadensis” in our study in is Rocky Mountain Elk (wapiti), which is inhabited at Minnesota. You can get some details in the reference “Masonbrink, R. E., Alt, D., Bayles, D. O., Boggiatto, P., Edwards, W., Tatum, F., Williams, J., Wilson-Welder, J., Zimin, A., Severin, A., & Olsen, S. (2021). A pseudomolecule assembly of the Rocky Mountain elk genome. PloS one, 16(4), e0249899. https://doi.org/10.1371/journal.pone.0249899”.

I think the samples in our study are accurate.

2.Page 1, lines 40-42: “The study based on mitochondrial genome and mitochondrial marker sequences show that Cervidae is divided into four subfamilies, Odocoileinae, Hydropotinae, Muntiacinae and Cervinae[3-5]” – the citations are not exact. Currently, there are only two modern monophyletic cervid subfamilies recognized: the subfamily Capreolinae (Odocouleinae is its junior synonym; this group is also known as telemetacarpal deer) and the subfamily Cervinae (or plesiometacarpal deer). Hydropotes is closely related to Capreolus and is included in the subfamily Capreolinae by all researchers. There are no debates concersisn the systematical position of Hydropotes in the current literature.

Thanks for your comment. We have corrected the errors expressed and move the sentence to discussion replaced the corresponding references in line 153-156, highlight with yellow background.

3.Page 2, line 50: "Cattle" should be capitalized.

Thanks for your comment. We have rewritten the introduction. And this sentence was removed.

4.Page 3-4, lines 120-121: “corresponding to Telemetacarpalia and Plesiometacarpalia, respectively, which also proposed by Brooke [23]” - better to move this sentence on correspondence of Capreolinae and Cervinae to Brooke's Telemetacarpalia and Plesiometacarpalia to the introduction chapter.

Thanks for your comment. We have reorganized the sentence and left it in the discussion in153-156, highlight with yellow background.

5.Page 4, lines 123-125: “However, previous studies have indicated that Hydropotes inermis is a distinct subfamily called Hydropotinae, which belongs to the Telemetacarpalia[24,25]” - The citations are not exact. Randi et al (1998) arrived to conclusion that Hydropotes is nested within the Odocoileinae, i.e. belongs to the subfamily Caprelinae. Gilbert et al (2006) included Hydropotes together with Capreolus to the tribe Capreolini within the subfamily Capreolinae.

Thanks for your comment. We have removed the citations.

6.Page 8, line 277: “Adaptive evolution in olfactory receptor gene families of eight Cervidae species were discovered” – The statement regarding the discovery of adaptive evolution in olfactory receptor gene families among eight Cervidae species requires further elaboration. It would be beneficial to include details such as the specific adaptations observed, the environmental conditions influencing these adaptations, the selection pressures involved, and the advantages conferred by these adaptations. Providing a summarized information and conclusion on these discoveries would enhance the clarity and impact of the study.

Thanks for your comment. “Adaptive evolution in olfactory receptor gene families of eight Cervidae species were discovered” is in the conclusion part as a summary of the Result 3.3 Gene family evolution. The details such as the environmental conditions influencing these adaptations, the selection pressures involved were elaborated in line 235-242, highlight with yellow background.

Reviewer 3 Report

Comments and Suggestions for Authors

This manuscript provides a hypothesis on the phylogenetic relationships and chromosome evolution within the family Cervidae using a phylogenomic approach.

The methods employed are adequate and the results should be considered for publication in my opinion. However, the manuscript also suffers from some issues that should be carefully addressed before it can be accepted for publication. 

The Abstract should be reworked and extended, including essential information on the methods used and the results obtained.

The Introduction is very short and feels quite incomplete. For example, given the research topic it would be very useful to specify something on the most supported phylogenetic trees already available and the controversy mentioned. Furthermore, it would also be informative to provide in this section some information on the karyotype diversity in Cervidae (e.g. range of chromosome number, chromosome morphology) (Fontana and Rubini 1990 Bio Systems, 24, 157–174; Proskuryakova et al. 2022 Cytogenetic and genome research, 162, 312–322.).

Concerning the Methods, it is unclear if the authors used protein or nucleotide data for phylogenetic inferences. Because nucleotide data should always be preferred in such analyses, if the authors used proteins they should motivate their choice.

In the Discussion, I am personally not sure if the presented phylogenetic tree is a better representation of the phylogenetic relationships of Cervidae, compared to that provided by Zhang and Zhang (2012). In fact, while I agree that the phylogenomic methods employed by the authors should be more accurate, the limited taxon sampling (compared to e.g. Zhang and Zhang 2012) may also negatively affect the results. The authors should mention this limitation when discussing the results.

Also in the Discussion, the authors should at least mention that in vertebrates, general differences at the chromosome level may be also due to differential heterochromatinization and transposable elements (see below).

Here below are some line-specific points.

Introduction

Line 34: “Chromosome karyotype” does not have a clear scientific meaning. Please change it to “Chromosomes” or “karyotypes”.

Line 38: Behavioral studies are not recommended in phylogenetic inferences. 

Line 40: Change “study” to plural if several distinct researches are cited.

Line 44-46. This period should be reformulated as it is quite unclear in its current form.

Materials and Methods

Line 69: Change “Phylogeny analysis” with “phylogenetic analysis”.

Lines 71-72: Please, better specify if nucleotide or protein (amino acid) data were used for phylogenetic inference. If protein data were used the authors should justify this choice.

Line 81: Outgroup selection is repeated from 2.1. Try to avoid similar repeated content.

Results and Discussion

Line 258: Please, at least mention that in vertebrates, general differences at the chromosome level may be also due to heterochromatinization and transposable elements (Mezzasalma et al. 2007 Salamandra 55(2):140-144; Petraccioli et al. 2007 Cytogenetic and Genome Research 157, 65-76).

Table 1. “9 mammalian species” is too generic, please be more specific , e.g. "9 Cervidae species used in the phylogenetic analysis”.

Under the “karyotype column” the authors should specify the ploidy as e.g. n = 33.

6).

Table 1. “9 mammalian species” is too generic, please be more specific , e.g. "9 Cervidae species used in the phylogenetic analysis”.

Under the “karyotype column” the authors should specify the ploidy as e.g. n = 33.

Comments on the Quality of English Language

The manuscript should be revised by a native speaker. There are a lot of quite confusing sentences, missing verbs, wrong singular/plural forms etc.

Author Response

The methods employed are adequate and the results should be considered for publication in my opinion. However, the manuscript also suffers from some issues that should be carefully addressed before it can be accepted for publication.

  1. The Abstract should be reworked and extended, including essential information on the methods used and the results obtained.

Thanks for your advice. The details of the experimental methods and results in the abstract have been expanded and rewritten in line 22-37, highlight with green background.

2.The Introduction is very short and feels quite incomplete. For example, given the research topic it would be very useful to specify something on the most supported phylogenetic trees already available and the controversy mentioned. Furthermore, it would also be informative to provide in this section some information on the karyotype diversity in Cervidae (e.g. range of chromosome number, chromosome morphology) (Fontana and Rubini 1990 Bio Systems, 24, 157–174; Proskuryakova et al. 2022 Cytogenetic and genome research, 162, 312–322.).

Thanks for your advice. We have expanded and rewritten the content of introduction in line 42-74, highlight with green background.

3.Concerning the Methods, it is unclear if the authors used protein or nucleotide data for phylogenetic inferences. Because nucleotide data should always be preferred in such analyses, if the authors used protein they should motivate their choice.

Thanks for your advice. We have rewritten 2.2 Phylogenetic analysis in method part and described the phylogenetic tree constructed using single-copy orthologous genes sequences. It is the nucleotide data, not protein data in line 88-96, highlight with green background. We are sorry for making you confuse.

4.In the Discussion, I am personally not sure if the presented phylogenetic tree is a better representation of the phylogenetic relationships of Cervidae, compared to that provided by Zhang and Zhang (2012). In fact, while I agree that the phylogenomic methods employed by the authors should be more accurate, the limited taxon sampling (compared to e.g. Zhang and Zhang 2012) may also negatively affect the results. The authors should mention this limitation when discussing the results.

Thanks for your advice. We have expanded the discussion of the limiting in samples in line 156-161, highlight with green background.

5.Also in the Discussion, the authors should at least mention that in vertebrates, general differences at the chromosome level may be also due to differential heterochromatinization and transposable elements (see below).

Thanks for your advice. I agree with you that the differences at the chromosome level may be also due to differential heterochromatinization and transposable elements in vertebrates. The chromosome-level genomes of nine species in our study do not have the annotation of transposable elements. And the analysis of phylogeny and chromosomal evolutionary events in our study do not involve the heterochromatinization and transposable elements. We think there is no need to discuss the differences at the chromosome level may be also due to differential heterochromatinization and transposable elements. I will take your comments into serious consideration. The study on transposable elements and heterochromatin of Cervidae will be our focus in the future.

Here below are some line-specific points.

Introduction

6.Line 34: “Chromosome karyotype” does not have a clear scientific meaning. Please change it to “Chromosomes” or “karyotypes”.

Thanks for your advice. We have changed “Chromosome karyotype” to “karyotype” in line 65, highlight with green background.

7.Line 38: Behavioral studies are not recommended in phylogenetic inferences.

Thanks for your advice. We have rewritten the introduction and removed the Behavioral study of Cervidae.

8.Line 40: Change “study” to plural if several distinct researches are cited

Thanks for your advice. We have rewritten the introduction and paid attention to the singular and plural of nouns in the article.

9.Line 44-46. This period should be reformulated as it is quite unclear in its current form.

Thanks for your advice. We have rewritten the content in line 67-69, highlight with green background.

Materials and Methods

10.Line 69: Change “Phylogeny analysis” with “phylogenetic analysis”.

Thanks for your advice. We have change “Phylogeny analysis” to “phylogenetic analysis” in line 87, highlight with green background.

11.Lines 71-72: Please, better specify if nucleotide or protein (amino acid) data were used for phylogenetic inference. If protein data were used the authors should justify this choice.

Thanks for your advice. We have rewritten 2.2 Phylogenetic analysis in method part and described the phylogenetic tree constructed using single-copy orthologous genes sequences. It is the nucleotide data, not protein data, in line 88-96, highlight with green background. We are sorry for making you confuse.

  1. Line 81: Outgroup selection is repeated from 2.1. Try to avoid similar repeated content.

Thanks for your comment. We have elaborated the outgroup selection in both the Materials and Methods part and the Results part, which we think is necessary and not a repetition of writing.  We hope to get your understanding.

Results and Discussion

13.Line 258: Please, at least mention that in vertebrates, general differences at the chromosome level may be also due to heterochromatinization and transposable elements (Mezzasalma et al. 2007 Salamandra 55(2):140-144; Petraccioli et al. 2007 Cytogenetic and Genome Research 157, 65-76).

Thanks for your advice. I agree with you that the differences at the chromosome level may be also due to differential heterochromatinization and transposable elements in vertebrates. The chromosome-level genomes of nine species in our study do not have the annotation of transposable elements. And the analysis of phylogeny and chromosomal evolutionary events in our study do not involve the heterochromatinization and transposable elements. We think there is no need to discuss the differences at the chromosome level may be also due to differential heterochromatinization and transposable elements. I will take your comments into serious consideration. The study on transposable elements and heterochromatin of Cervidae will be our focus in the future.

14.Table 1. “9 mammalian species” is too generic, please be more specific , e.g. "9 Cervidae species used in the phylogenetic analysis”.

Thanks for your advice.

The table not only contain Cervidae species, but also an outgroup species, cattle. And the genome of 9 species not only used in the phylogenetic analysis, it is also used for chromosome evolution of Cervidae. We have changed the name to “the genomic information of 9 species used in our study” in line 97, highlight with green background.

15.Under the “karyotype column” the authors should specify the ploidy as e.g. n = 33.

6).

Thanks for your advice, we have change “karyotype” to “Haplotype”, in Table 1, highlight with green background.

16.Table 1. “9 mammalian species” is too generic, please be more specific , e.g. "9 Cervidae species used in the phylogenetic analysis”.

Thanks for your advice.

The table not only contain Cervidae species, but also an outgroup species, cattle. And the genome of 9 species not only used in the phylogenetic analysis, it is also used for chromosome evolution of Cervidae. We have changed the name to “the genomic information of 9 species used in our study” in line 97, highlight with green background.

17.Under the “karyotype column” the authors should specify the ploidy as e.g. n = 33.

Thanks for your advice, we have change “karyotype” to “Haplotype”, in Table 1, highlight with green background.

Round 2

Reviewer 2 Report

Comments and Suggestions for Authors

The taxonomy of red deer Cervus elaphus and wapiti Cervus canadensis indeed it very confused because generally taxonomy was a disregarded biological discipline and many researchers, especially in genetics, always have had a superficial approach to taxonomy and systematics of species that they analyzed. This is not a critic to the authors, this is a generally regrettable situation and the authors of the paper that I am reviewing try to struggle with this situation.

Tarym deer is usually considered as one of the most primitive subspecies of red deer Cervus elaphus. In all genetical studies cited in the paper Cervus elaphus and Cervus canadensis were considered just as groups of subspecies and they were lumped together in one species Cervus elaphus. From these confusions in the literature, the paragraph on the page 2, lines 47-62 is also quite confused. Please, take a time to read the taxonomical study of Cervus canadensis and Cervus elaphus and ecological and evolutionary relationships between them:

  1. Croitor, R. & Obada, Th. 2018. On the presence of Late Pleistocene wapiti, Cervus canadensis Erxleben, 1777 (Cervidae, Mammalia) in the Palaeolithic site Climăuți II (Moldova). Contributions to Zoology, 86 (4): 273-296.
  2. Croitor, R., 2020. A new form of wapiti Cervus canadensis Erxleben, 1777 (Cervidae, Mammalia) from the Late Pleistocene of France. Palaeoworld, 29 (4): 789-806.

I hope these publications will help to understant the taxonomical confusions of wapiti and red deer.

Author Response

The taxonomy of red deer Cervus elaphus and wapiti Cervus canadensis indeed it very confused because generally taxonomy was a disregarded biological discipline and many researchers, especially in genetics, always have had a superficial approach to taxonomy and systematics of species that they analyzed. This is not a critic to the authors, this is a generally regrettable situation and the authors of the paper that I am reviewing try to struggle with this situation.

Tarym deer is usually considered as one of the most primitive subspecies of red deer Cervus elaphus. In all genetical studies cited in the paper Cervus elaphus and Cervus canadensis were considered just as groups of subspecies and they were lumped together in one species Cervus elaphus. From these confusions in the literature, the paragraph on the page 2, lines 47-62 is also quite confused. Please, take a time to read the taxonomical study of Cervus canadensis and Cervus elaphus and ecological and evolutionary relationships between them:

Croitor, R. & Obada, Th. 2018. On the presence of Late Pleistocene wapiti, Cervus canadensis Erxleben, 1777 (Cervidae, Mammalia) in the Palaeolithic site Climăuți II (Moldova). Contributions to Zoology, 86 (4): 273-296.

Croitor, R., 2020. A new form of wapiti Cervus canadensis Erxleben, 1777 (Cervidae, Mammalia) from the Late Pleistocene of France. Palaeoworld, 29 (4): 789-806.

I hope these publications will help to understant the taxonomical confusions of wapiti and red deer.

Thanks for your sincere comment. We are sorry for making you confuse. We have carefully polished this part in line 46-61, highlight with cyan background, which we want to express is red deer (Cervus elaphus), Tarim red deer (Cervus hanglu) and wapiti (Cervus canadensis) have been classified as three separate species based on current study (Lorenzini, R.; Garofalo, L. Insights into the evolutionary history of Cervus (Cervidae, tribe Cervini) based on Bayesian analysis of mitochondrial marker sequences, with first indications for a new species. Journal of Zoological Systematics and Evolutionary Research 2015, 53, 340-349). And the phylogenetic relationship among sika deer, red deer, wapiti and Tarim red deer can not reach a consensus. Our study mainly focused on the phylogenetic relationship of eight Cervidae species through whole genome comparison. And we know that classification is close related to phylogeny. Thanks for the references you have provided. We have read some parts and felt difficult in reading these references. It must take us a lot of time to fully understand these references.

Reviewer 3 Report

Comments and Suggestions for Authors

The authors improved different sections of the manuscript, which can now be accepted for publication.

There are still some minor issues the authors should carefully check.

In table 1 the use of "Haplotype" is incorrect. I think the authors made some confusion with "haploid karyotype" which should be used instead and has a very different meaning.

The authors decided to not include a broader explanation of possible chromosomal evolutionary dynamics in Cervidae, leaving in the Discussion a very limited interpretation of the results which could therefore be extended and made more interesting for potential readers.  

Comments on the Quality of English Language

A spell check is still required, along with the reorganization of some sentences

Author Response

The authors improved different sections of the manuscript, which can now be accepted for publication.

There are still some minor issues the authors should carefully check.

1.In table 1 the use of "Haplotype" is incorrect. I think the authors made some confusion with "haploid karyotype" which should be used instead and has a very different meaning.

Thanks for your advice, we have corrected "Haplotype" to "haploid karyotype", in table 1, highlight with red background.

2.The authors decided to not include a broader explanation of possible chromosomal evolutionary dynamics in Cervidae, leaving in the Discussion a very limited interpretation of the results which could therefore be extended and made more interesting for potential readers. 

Thanks for your advice. The reason why we do not give a broader explanation of possible chromosomal evolutionary dynamics is that Cervidae is a vast family with nearly 55 species, but the chromosomal-level genomes covered in our article only include 7 species, thus representing a very limited range. We do not believe that the content of our study can illustrate such complexity. As our title suggests, we have only explored chromosomal evolutionary events in 8 species and have discussed these in conjunction with existing cytogenetic research. We have endeavored to elucidate the chromosomal evolutionary dynamics in Cervidae. In discussion section, we have discussed the possible chromosomal evolutionary pattern in four species of Cervusin line 277-285, highlight with red background.

We have polished the article carefully.
